# Synthesis, Formulation and Characterization of Immunotherapeutic Glycosylated Dendrimer/cGAMP Complexes for CD206 Targeted Delivery to M2 Macrophages in Cold Tumors

**DOI:** 10.3390/pharmaceutics14091883

**Published:** 2022-09-06

**Authors:** Marija Petrovic, Alexandre Porcello, Stoyan Tankov, Oliwia Majchrzak, Martin Kiening, Annick Clara Laingoniaina, Tayeb Jbilou, Paul R. Walker, Gerrit Borchard, Olivier Jordan

**Affiliations:** 1School of Pharmaceutical Sciences, University of Geneva, 1206 Geneva, Switzerland; 2Institute of Pharmaceutical Sciences of Western Switzerland, University of Geneva, 1206 Geneva, Switzerland; 3Translational Research Centre in Oncohaematology, University of Geneva, 1206 Geneva, Switzerland

**Keywords:** PAMAM, cGAMP, CD206, M2 macrophage-targeted delivery, nanoparticle orthogonal characterization

## Abstract

Anti-tumor responses can be achieved via the stimulation of the immune system, a therapeutic approach called cancer immunotherapy. Many solid tumor types are characterized by the presence of immune-suppressive tumor-associated macrophage (TAMs) cells within the tumor microenvironment (TME). Moreover, TAM infiltration is strongly associated with poor survival in solid cancer patients and hence a low responsiveness to cancer immunotherapy. Therefore, 2′3′ Cyclic GMP-AMP (2′3′ cGAMP) was employed for its ability to shift macrophages from pro-tumoral M2-like macrophages (TAM) to anti-tumoral M1. However, cGAMP transfection within macrophages is limited by the molecule’s negative charge, poor stability and lack of targeting. To circumvent these barriers, we designed nanocarriers based on poly(amidoamine) dendrimers (PAMAM) grafted with D-glucuronic acid (Glu) for M2 mannose-mediated endocytosis. Two carriers were synthesized based on different dendrimers and complexed with cGAMP at different ratios. Orthogonal techniques were employed for synthesis (NMR, ninhydrin, and gravimetry), size (DLS, NTA, and AF4-DLS), charge (DLS and NTA), complexation (HPLC-UV and AF4-UV) and biocompatibility and toxicity (primary cells and hen egg chorioallantoic membrane model) evaluations in order to evaluate the best cGAMP carrier. The best formulation was selected for its low toxicity, biocompatibility, monodispersed distribution, affinity towards CD206 and ability to increase M1 (STAT1 and NOS2) and decrease M2 marker (MRC1) expression in macrophages.

## 1. Introduction

A hot tumor microenvironment (TME) is characterized by immune cell infiltration, consisting mainly of antigen-presenting cells (APCs) (mainly dendritic cell (DC) pro-inflammatory M1 subtype macrophages), T cells, and NK cells. These characteristics lead to responsiveness to immune therapy (particularly to immune checkpoint blockade) and overall better prognosis and patient survival. In contrast, a cold TME either lacks immune infiltration or is infiltrated by predominantly immunosuppressive tumor-associated macrophages (TAMs), which often exhibit an M2-like pro-tumor phenotype.

Due to uncontrolled growth and aberrant microvasculature, there is increased oxygen consumption with decreased supply in the TME, which leads to the formation of hypoxic regions. When macrophages are recruited in hypoxic tumor regions, their polarization can be altered towards an M2-like pro-tumor phenotype via the activity of hypoxia-induced factors. Therefore, shifting the phenotype from pro-tumoral M2 to anti-tumoral M1 macrophages is an attractive strategy for many solid tumors that are highly infiltrated with TAMs, such as prostate cancer, pancreatic cancer, and glioblastoma, with the aim to increase the immune infiltration of anti-tumoral cells within the TME [1,2].

A series of recent studies have indicated that STING pathway activation within macrophages results in a phenotype shift from the M2 to M1 polarization phenotype [3,4]. Moreover, in APC (both macrophages and DC), stimulator of interferon genes (STING) pathway activation causes increased type I interferon (IFN) production, facilitating the efficient presentation of tumor antigens and priming of CD8+ cytotoxic T cells [5]. The STING pathway is an innate immunity pathway activated by pathogen-derived or tumoral dsDNA or directly by STING ligands. STING ligands were either synthesized or extracted from eukaryotes/prokaryotes. As it was shown that eukaryotic cyclic dinucleotide (CDN)—2′3′ Cyclic GMP-AMP (2′3′ cGAMP)—showed a higher affinity towards the STING pathway compared to prokaryotic 3′3′ cGAMP, for the purpose of this work, 2′3′ cGAMP was the ligand of choice [6].

2′3′ cGAMP is a negatively charged dinucleotide due its phosphate groups; it therefore has limited transfection efficiency owning to the negative charge of cell membranes. Previous research has shown that intratumor (i.t.) administration is preferred compared to the intravenous (i.v.) route, due to the presence of enzymes in the blood stream, which would eventually degrade cGAMP [1]. Moreover, the only STING ligand (chemically modified cGAMP-ADU-S100) currently tested in clinical trials is administered i.t. (NCT02675439, NCT03172936 and NCT03937141). Therefore, in order to successfully transfect APCs with 2′3′ cGAMP, there is demand for a positively charged carrier [1]. Several carriers for STING ligands, presenting different interesting characteristics, are currently available. Liposomes provide optimal drug protection due to their low immunogenicity and high biocompatibility [1]. Moreover, the blood half-life of liposome–drug complexes can be prolonged and their kidney excretion lowered. The disadvantages of liposomes as drug carrier systems include their low complexation efficiency (CE) and their biological and colloidal instability [1]. Drug protection from degradation and ensuring a controlled and targeted delivery are the main advantages of hydrogels. Nonetheless, hydrogels tend to release the drug before reaching the target site, due to their high-water content. This leads to a burst reaction and the early release might potentially cause toxic effects. Polymers’ properties are critical to ensure appropriate transfection. Cationic polymers can protect genetic material and ensure a targeted delivery. Additionally, they are chemically easily modified in terms of surface charge, molecular weight, and architecture. Therefore, we chose polyamidoamine (PAMAM), a positively charged polymer, as our drug delivery platform. Indeed, this polymer is the first dendrimer to be described [7], and has been widely used over the last 20 years for gene and drug delivery [8].

PAMAM dendrimers are suitable for condensing and thus carrying drugs and genes thanks to their great number of branches capped with primary amino groups, high drug loading capacity, and tunable size in the nanometer range [9,10]. Based on the number of amino groups on their surface, PAMAM dendrimers are divided into different generations (G 0-9) [8]. The higher the dendrimer generation (with a higher number of amino groups), the denser the packing of the defined structure. However, toxicity remains one of the main issues when working with PAMAM polymers. Up to now, G1–G5 are mostly used for gene therapy considering their efficacy/toxicity balance [8,9,11,12]. Surface engineering is a common strategy used to curb PAMAM cytotoxicity. This can consist of coating and thus blocking the surface-reactive amines with a given molecule or molecular group hindering the formation of cytotoxic species [10,13]. Importantly, these groups can be used to recognize specific cells, conferring targeted properties to the nanovector.

A series of recent studies have indicated that STING pathway activation within macrophages results in a phenotype shift from the M2 to M1 polarization status [3,4]. Therefore, 2′3′ cGAMP delivery to M2 macrophages is one of the main goals of this study. It is important to highlight differences in macrophage phenotype, such as gene signatures for classically activated M1 macrophages—*Il1b*, *Nos2* and *Stat1*—and for alternatively activated M2 macrophages, such as *Mrc1*, which codes CD206, *Arg1* and *Cdh1* [14,15]. Moreover, the surface expression of macrophage activation/differentiation markers is crucial for the targeting strategy. CD68, CD80 and CD86, are prototypical M1 markers, and CD206 is widely accepted as an M2 marker [16]. CD206 is a mannose receptor, which mediates pathogen-associated molecular pattern (PAMP)/damage-associated molecular pattern (DAMP) recognition [17,18]. This receptor triggers the uptake to the endolysosomal compartments for presentation to T-cells [19]. High expression levels of CD206 in TAMs are generally associated with poor prognosis [20,21]. Sugars such as fucose, mannose or glucuronic acid are ligands of this receptor, and they are natural components of the bacterial wall [22]. Since one of the ways to polarize macrophages within TME is the activation of the STING pathway, this study aimed to deliver 2′3′ cGAMP to TAMs via sugar-coated PAMAM, mimicking bacteria. To achieve this, D-glucuronic acid (Glu) was chosen due to its affinity towards CD206 and the ability of its carboxylic group to covalently bind to surface PAMAM’s primary amino groups [22,23]. By delivering cGAMP to the macrophage cytosol, we expect a shift in surface markers’ expressions and gene panel activation.

Unlike other studies, which encapsulate cGAMP to trigger immune responses within the TME, here we synthesized a carrier targeting M2 macrophages, which will additionally boost macrophage polarization [1]. Besides the M2 targeting approach, this study developed a formulation of nanocomplexes (NCs) out of 2′3′ cGAMP and PAMAM-Glu, which were characterized in depth using orthogonal approaches. Their in vitro activities (based on the gene expression of macrophages and their surface markers) and in vivo biocompatibility (chicken embryos) were fully evaluated.

## 2. Materials and Methods

### 2.1. Chemicals

PAMAM with a 3rd generation (G3) and 4th generation (G4) ethylenediamine cores (20 wt% and 10 wt% solution in methanol, respectively) were purchased from Sigma-Aldrich (St. Louis, MO, USA) D-Glucuronic acid (D Glu) from Alfa Aesar (Karlsruhe, Germany), and 2′3′-cGAMP was obtained from Invivogen (Toulouse, France). PAMAM-D glu is referred to in the following text as either PG3 or PG4 depending on whether it was synthesized using the above-mentioned G3 or G4 PAMAM.

### 2.2. Synthesis of G3 and G4 PAMAM-D-Glucuronic Acid Derivatives

PAMAM G3 (20 wt% solution in methanol) or PAMAM G4 (10 wt% solution in methanol) was transferred into a round-bottom flask to evaporate the methanol using a rotary evaporator (337 mmHg at 40 °C for 20 min). In the same round-bottom flask, distilled water was added (10 times the initial volume) and dry PAMAM was dissolved under magnetic stirring. The solution was then lyophilized (Freeze Dryer Alpha 1–4 LDplus, Christ, Osterode am Harz, Germany) (48 h, 1.5 × 10^−1^ mbar, −80 °C) before storage at 4 °C. Lyophilized PAMAM G3 or G4 was dissolved in distilled water at 0.1% (*w*/*v*) under magnetic stirring. 1-Ethyl-3-(3-dimethylaminopropyl) carbodiimide (EDC) (5 eq. NH_2_), N-hydroxysuccinimide sodium salt (NHS) (2 eq. NH_2_) and D-Glucuronic (0.1 eq. NH_2_ for PG3 0.1 eq. NH_2_ for PG4 0.1 eq., and 0.3 eq NH_2_ for PG4 0.3 eq.) were added. pH was adjusted to 5.5 (827 pH Lab Meter from Metrohm) using NaOH/HCl. The reaction was allowed to proceed for 12 h under stirring at room temperature. The reaction mixture (G3 or G4 PAMAM-D-glucuronic acid) was dialyzed three times against 5% (*w*/*v*) NaCl (1000 MWCO, 4 h, room temperature) and three times against distilled water (1000 MWCO, 4 h, room temperature) and then lyophilized before storage at 4 °C. 

### 2.3. Physico-Chemical Characterization of PG3, PG4 and PG/cGAMP

#### 2.3.1. ^1^H NMR

Chemical structures and degrees of substitution (DS) were assessed by ^1^H NMR spectroscopy on a Bruker Avance Neo 600 MHz NMR spectrometer using D2O as a solvent [24,25,26]. DS was measured using two characteristic peaks—one derived from the dendrimer structure (ethylenediamine core: ~2.56 ppm; 4 protons), the other from the hydrogen C2 (H18: ~3.5 ppm; 1 proton [27]; Appendix A), which do not overlap in the spectrum of the product. The disappearance of the glu carboxyl group was used to determine if grafting was successful on PG3 or PG4. DS was defined as the ratio of the hydroxyl to ethylediamine protons, as measured by peak area.

#### 2.3.2. Ninhydrin Assay

DS were confirmed via UV-vis spectroscopy using ninhydrin reagent. Ninhydrin assay was performed as previously described [28]. The colorimetric assay was performed in sodium acetate buffer (pH 5.4) using ninhydrin (2, 2-dihydroxyindane-1, 3-dione) reagent solution to detect free primary amino groups. The absorbance was measured at 570 nm. The DS were calculated from an absorbance calibration curve measured at concentrations of 1, 0.75, 0.5, 0.25, 0 mg/mL in in Milli-Q^®^ water.

#### 2.3.3. Rheology

The rheological behaviors of PG4 0.1 eq. and PG3 0.1 eq. at different concentrations (% *w*/*v*) in Milli-Q^®^ water were determined on a HAAKE Mars Rheometer™ (Thermo Scientific, Waltham, MA, USA) equipped with a cone-plate C35 2°/Ti measuring geometry on 420 µL samples. The complex viscosity (η*) was assessed at 25 °C and a constant oscillatory frequency of 1 Hz. Shear stress was set to 1 N/m^2^ in all experiments to stay in the linear viscoelastic region (LVE). Additional assays were performed with a frequency sweep from 0.1 Hz to 10 Hz. The assays were performed in triplicate on the day following the PG4 0.1 eq. and PG3 0.1 eq. solubilization. 

#### 2.3.4. Karl-Fischer

The relative remaining moisture level were assessed by the Karl Fischer method (Coulometric KF Titrator Compact C30SD, Mettler Toledo, Greifensee, Switzerland) on ≥10 mg samples.

#### 2.3.5. Dynamic Light Scattering (DLS)

Z-average, polydispersity index (PDI) and zeta potential of PAMAM alone and PG/cGAMP NCs were measured by DLS and ELS (Zetasizer nano-ZS, Malvern Panalytical, Malvern, UK, 633 nm He–Ne laser and 173° scattering angle; temperature: 25 °C; refractive index corresponding to water (RI): 1.331; 0.8872 cP, attenuator adjusted automatically). Size measurements were performed in ZEN0040 disposable cuvettes and zeta potential was measured using DTS1070 reusable cuvettes. Data were analyzed by Malvern Instruments Zetasizer Software version 7.13. Particle z-average, PDI and zeta potential were determined in Milli-Q^®^ water (*n* = 9), first for a PAMAM characterization alone at 0.1 mg/mL. Once NCs were prepared, they were measured at concentration 0.01 mg/mL of cGAMP in PG/cGAMP NC. The results are presented as mean ± SD of triplicate measurements of three different batches of PG3 and PG4.

#### 2.3.6. Nanoparticle Tracking Analysis (NTA)

The size and zeta potential of PG/cGAMP NCs were analyzed at 0.01 mg/mL in Milli-Q^®^ water (*n* = 3), which corresponds to cGAMP in PG/cGAMP NC, by nanoparticle tracking analysis (NTA; Particle Matrix ZetaView, laser wavelength 520 nm, ZP at pulses −20 and 20 V, measurements were taken over 1 cycle at 11). The cell used was ZetaView^®^ NTA—ZNTA. The results are presented as mean ± SD of measurements of three different batches of PG3 and PG4.

#### 2.3.7. Asymmetric Flow Field Flow Fractionation (AF4)-RI-UV-MALS-DLS

To determine the recovery of cGAMP (%), radius of gyration (Rg), hydrodynamic radius (Rh), shape factor (Rg/Rh) and molecular weight (MW) of the PAMAM, PG and PG/cGAMP NC, we performed AF4-UV-MALS-DLS analysis (Postnova, NovaFFF AF2000; UV detector—Waters 2487 Dual lambda absorbance detector, DLS, Malvern Zetasizer nano ZS). We used a new membrane (amphiphilic regenerated cellulose) that is suitable for cationic polymers with a 10 kDa molecular weight cut-off and a 350 µm spacer. The mobile phase was 0.05% Novachem (Postnova, NovaChem Surfactant). The particle size or gyration radius (Rg) and molecular weight were determined by multiangle light scattering (MALS), using Berry formalism, level 2, where linearity was obtained once 28° and 156° detectors were excluded. The hydrodynamic radius (Rh) or z-average was determined by AF4-DLS.

PG/cGAMP NCs were analyzed at a concentration of 0.1 mg/mL of cGAMP prepared in Milli-Q^®^ water. An amount of 50 µL of sample (5 µg) was manually injected. The method was optimized based on the Postnova^®^ latex method for a duration of 25 min (Table 1). Malvern Zetasizer software was set to “flow” measurement and the data were exported to Postnova software for analysis. All raw data corresponding to Rh, Rg and Rg/Rh for the selected eluted peak were exported to an Excel file and calculated as average value of all PG/cGAMP NC examined within the peak range selected. The recovery of cGAMP with cross flow from PG/cGAMP complexes (%) was calculated according to Equation (1): (1)$$\mathrm{cf}\;\left(\%\right)=\frac{\mathrm{cf}\;\mathrm{experimental}\;\mathrm{cGAMP}\;\mathrm{conc}}{\mathrm{theoretical}\;\mathrm{cGAMP}\;\mathrm{conc}}\times 100$$
where Rcf stands for recovery with cross flow applied.

#### 2.3.8. Gravimetry

Three different batches of PG3 0.1 eq., PG4 0.1 eq. and PG4 0.3 eq. at 1.5 mg/mL in water were prepared. Samples were filtered over 0.45 µm membrane (CHROMAFIL Xtra PA-20/13, Macherey-Nagel GmbH & Co, KG, Duren, Germany) and lyophilized (Christ alpha 2–4 ld plus, Osterode am Harz, Germany). Recovery (%) and concentration (mg/mL) before and after filtration were investigated. 

#### 2.3.9. Light Microscopy

PG3 and PG4 of different concentrations, from 0.05 mg/mL to 1 mg/mL, were placed on microscope slides (76 × 26 mm, Asia Premium, Braunschweig, Germany), covered by cover slips (22 × 22 mm, Menzel glaser, Braunschweig, Germany) and analyzed in water by light microscopy (ZEISS, Oberkochen, Germany). Microscopy was set at 10× magnification with phase contrast 10×.

#### 2.3.10. Scanning Electron Microscopy (SEM)

For imaging, 5 µL of 0.01 mg/mL PG/cGAMP NC in Milli-Q^®^ water was placed on a copper grid and then vacuum-dehydrated on silica gel overnight. Samples were sputter-coated with 20 nm gold (Leica EM SCD500, Wetzlar, Germany) and imaged by scanning electron microscopy (SEM, JSM-7001F, JEOL, Tokyo, Japan).

#### 2.3.11. UPLC-UV for CE% Determination

PG/cGAMP NCs were separated from free cGAMP by centrifugation using VIVASPIN 3000 MWCO (Littleton, MA, USA) at 4000× *g* over 20 min and the filtrate was analyzed [29]. cGAMP loading was quantified by a Waters UPLC system (Milford, MA, USA), consisting of an ACQUITY UPLC System, and an ACQUITY BEH C18 column (1.7 µm, 2.1 × 50 mm). The mobile phases ammonium formate (pH 7.4, 10 mM) and acetonitrile (pH 7.4) were used with a gradient starting at 100:0 for 0.2 min, 1.4 min 90:10 at 1.4 min, 10:90 at 2 min, and 100:0 at 2.26 min, kept for 3.2 min. Analytes were detected and quantified by ACQUITY UPLC PDA at a wavelength of 256 nm. The flow rate was fixed at 0.3 mL/min. PEI/cGAMP NCs were prepared in PBS at a concentration of 0.01 mg/mL PEI/cGAMP (2.2), with 5 µL injection volume. Results are presented as mean ± SD of three different batches experiments (*n* = 3). Complexation efficiency (CE) was calculated as shown in Equation (2):(2)$$\mathrm{CE}\;\left(\%\right)=100-\frac{\mathrm{experimental}\;\mathrm{cGAMP}\;\mathrm{conc}}{\mathrm{theoretical}\;\mathrm{cGAMP}\;\mathrm{conc}}\times 100$$

#### 2.3.12. Fabrication of PG/cGAMP NCs

A stock of cGAMP in LAL water (endotoxin-free water) was kept at 1 mg/mL at −20 °C. PG (PG3 and PG4) were freshly dissolved in water every time before complexation with cGAMP. Two N/P (PG/cGAMP NCs) ratios of 1/1 and 2/1 were used. The concentrations of cGAMP in PG/cGAMP NC suspensions were fixed at 0.01 mg/mL for physicochemical characterization, 0.25 mg/mL (5 µg per well) for in vitro, and 5 µg and 10 µg for in vivo experiments.

As an example calculation, to obtain 0.010 mg/mL cGAMP concentration in PG/cGAMP NC, we added in equal parts a solution of 0.020 mg/mL of cGAMP and a solution of either 0.0125 mg/mL (ratio 1/1) or 0.0250 mg/mL (ratio 2/1) PAMAM. PG/cGAMP NCs were prepared by an ionic gelation method, as described previously for chitosan particles. PG was added dropwise to the cGAMP solution while vortexing for 30 s. Complexation was completed at room temperature (RT) over 30 min. At pH 6, both PG and cGAMP were fully ionized (chemicalize.com, accessed on 11 November 2017).

PG/cGAMP NCs are labelled as either PG3 1/1, PG3 2/1 or PG4 1/1, PG4 2/1, depending on the polymer used as a carrier.

#### 2.3.13. MST—kD Determination

A dye—Monolith His-Tag Labeling Kit RED-tris-NTA 2nd Generation (MO-L018) (NanoTemper Technologies, Munich, Germany)—was suspended at 5 µM in 1 × 20 mM HEPES (pH 7.4) supplemented with 0.1% pluronic F127. All further dilutions were also prepared in 1 × 20 mM HEPES (pH 7.4) supplemented with 0.1% pluronic F127.

Dye: 90 µL of 100 nM and 90 µL of 16 nM Recombinant Mouse Mrc1 (CD206) protein (Leu225 Ser492, His and T7-tagged #Mrc1-7099M Creative BioMart^®^), where the fluorescent signals of CD206 correspond to the typical detection limit of the Monolith NT.115 instrument (NanoTemper Technologies, Munich, Germany). The mixture was incubated for 30 min at room temperature. After incubation, samples were centrifuged at 4 °C, at a speed of 15,000× *g*, and the supernatant was transferred to new PCR tubes. The final concentration of the fluorescently labelled CD206 protein was 4 nM. A total of 10 µL of ligand—glucuronic acid (and corresponding PG3 and PG4 concentrations)—was prepared in 16-tube serial 64—0.001953 M dilution. In each tube, 10 µL of previously prepared His-tagged CD206 protein was added to the ligand to a final volume of 20 μL using low-bind pipette tips. The samples were loaded into premium treated capillaries using capillary force. Three sets of six repetitions were performed on the NanoTemper^®^ Monolith NT.115 (NanoTemper Technologies GmbH, Munich, Germany) in premium capillaries (MO-K025) at 25 °C using 40% MST power and 100% light-emitting diode (LED) with laser off/on times of 5 s and 30 s. The system was operated with the latest version of the MO control software (v1.6) with data analysis performed using the NanoTemper^®^ analysis software. Results are presented as mean ± SD (*n* = 18; 6 independent experiments in triplicate for three different laser power values).

### 2.4. In Vitro Experimentation

Bone marrow-derived dendritic cells (BMDC) were differentiated from C57BL/6JRj mice bone marrow cells by Dr. Julia Wagner. Cells were generated as previously described, with the only difference being that cells were split (1 to 2) at day 3. On day 6, BMDCs were seeded on a 96-well flat-bottom plate (Greiner) at a density of 1 × 10^5^ cells/well and stimulated with different concentrations of the previously described formulations [30,31]. BMDC were cultured in T75 Corning culture flasks with vented caps at 37 °C in 5% CO_2_ in RPMI 1640 cell culture medium, 10% fetal bovine serum (FBS, Gibco), supplemented with 1% Pen/Strep, 1% L-glutamine, 0.1% b-mercaptoethanol (50 mM) and 0.5% sodium pyruvate. Bone marrow cells were extracted from the femurs of C57BL/6J mice (Charles River) under license GE/33/20. The cells were treated with ACK lysis buffer for 3 min (Gibco) to remove red blood cells. Cells were then centrifuged, resuspended and cultured in RPMI 1640 medium (Gibco) supplemented with 10% fetal bovine serum (Sigma), 1% antibiotic solution (10,000 units per mL penicillin and 10,000 μg per mL streptomycin) (Gibco), MEM NEAA (Gibco), 10 mM HEPES (Gibco), 1 mM Sodium Pyruvate (Gibco), 50 µM 2-Mercaptoethanol (Gibco) and 10 ng/mL M-CSF (Immunotools) to induce differentiation into bone marrow-derived macrophages (BMDMs). The cells were cultured in two groups (M1 and M2) at 37 °C and 5% CO_2_. After 72 h post extraction, the original medium in all BMDM groups was aspirated to remove non-adherent cells and fresh medium was added to continue culturing. For the M1 group on day 5, IFN-γ (20 ng/mL) (Immunotools) was added for 1 h, followed by lipopolysaccharide (LPS) for 48 h (100 ng/mL). For the M2 group at day 5, IL-4 (20 ng/mL) (Immunotools) and IL-13 (20 ng/mL) (Immunotools) were added. On day 7 of culture, all the BMDMs were detached using TryLE Express buffer (Gibco) and the cells were subcultured on sterile 96-well cell-repellent plates (Cellstar).

#### 2.4.1. Interferon β Level

BMDCs and M1 or M2 BMDMs were exposed to the different NCs (1/1 and 2/1 PG/cGAMP complexes made of PG3 and PG4) at 0.25 mg/mL (5 µg) cGAMP in formulation in half 0.9% NaCl and half respective cell culture media in a 96-well plate. After 24 h, supernatants were removed and assessed for Interferon β (IFN β) content and cells used for further flow cytometry measuring CD80/86. The protocol was as follows:A 96-well plate was coated with 1/625 diluted 50 µL/well anti-IFN β, RMMB-1 (PBL, 22400-1) and incubated at 4 °C overnight;The next day, standard dilutions of mouse 1/840 dilution (from 1000 U/mL–7.8 U/mL) of IFN β (PBL, 12400-1) and samples were added (total 50 µL/well);Two hours later, the detection antibody, a 1/12,169 dilution of anti-IFN β (PBL, 32400-1), was added;Following this, the secondary antibody, anti-rabbit IgG, 1/1000 dilution HRP-linked (Cell Signaling, 7074S), was added;Reaction was induced by the addition of TMB substrate (BD OptEIA, 555214);It was stopped by the addition of 25 µL H_2_SO_4_;Optical absorbance was measured at 450 and 570 nm using a microplate reader (Biotek SynergyMx) and IFN beta levels are expressed as U/mL.

Results are presented as mean ± SD of triplicate experiments (*n* = 9; 3 wells per experiment 3×).

#### 2.4.2. Flow Cytometry Analysis

For flow analysis, 1 × 10^5^ M1, M2 or M0 BMDMs cells were used. After seven days, polarized BMDMs were incubated for 24 h on 96-well cell-repellent plates (Cellstar) with the corresponding treatment (PG3, PG4, cGAMP and NCs). After 24 h, cells were collected and washed with 1 × PBS followed by blocking with CD16/CD32 (Mouse Fc Block, Clone 2.4G2) for 15 min at 4 °C; they were then incubated with antibodies to obtain the proteins of interest (30 min, 4 °C), CD206-FITC, CD68-BV605, CD86 APC/Fire750, CD80-APC (Biolegend). 

CD80-Pe (PE conjugated anti-mouse CD80 Antibody Biolegend);CD86-Bv510 (Brilliant Violet 510™ conjugated anti-mouse CD86 Antibody, Biolegend);Live/Dead Fixable Violet Dead Cell Stain (ThermoFisher)CD 206-FITC (FITC conjugated anti-mouse CD206, Biolegend);CD68-BV605 (Brilliant Violet 605 anti-mouse CD68, Biolegend).

Unstained cells were used to establish the flow cytometer settings. Fluorescence bead controls (VersaComp Antibody Capture Kit, Beckman Coulter) were used for compensation. Flow cytometry data were acquired on a Gallios flow cytometer (Beckman Coulter), and the data were analyzed using Kaluza Software.

After incubation, cells were washed with FACS buffer (1× PBS, 2% FBS and 0.55 g/L Na-azide (*w*/*v*)), and finally resuspended in 100 µL FACS buffer. Results are presented as mean ± SD of triplicate experiments (*n* = 3).

#### 2.4.3. Quantitative RT-PCR Assays

The BMDMs were seeded on 96-well cell-repellent plates for 24 h.

cDNA was synthesized from 0.5 µg of total RNA using a mix of random hexamers—oligo d(T) primers and PrimerScript reverse transcriptase enzyme (Takara bio inc. Kit, Tokyo, Japan)—following the supplier’s instructions. Sybr green assays were designed using the program Primer Express v 2.0 (Applied Biosystems, Waltham, MA, USA) with default parameters. Amplicon sequences were aligned against the mouse genome by BLAST to ensure that they were specific to the genes used. Arg1 (forward: GCAGAGGTCCAGAAGAATGG; reverse: AGCATCCACCCAAATGACAC), Mrc1 (forward: TGTCAACCCTGCAGATTTCAAG; reverse: GAGTGGCTTACGTGGTTGTTTC), Cdh1 (forward: GAGCGTGCCCCAGTATCGT; reverse: GGCTGCCTTCAGGTTTTCATC), Il1b (forward: ACCCTGCAGCTGGAGAGTGT; reverse: CCATCTTCTTCTTTGGGTATTGCTT), Nos2 (forward: CCGGCAAACCCAAGGTCT; reverse: CCGTGGAGCACGCTGAGTA) and Stat1 (forward: GCCTGGATCAGCTGCAAAG; reverse: GCTGCAGGGTCTCTGCAAC). Expression was calculated by normalization with Gapdh (forward: TCCATGACAACTTTGGCATTG; reverse: CAGTCTTCTGGGTGGCAGTGA) and Eefa1a1 (forward: TCCACTTGGTCGCTTTGCT; reverse: CTTCTTGTCCACAGCTTTGATGA). Oligonucleotides were obtained from Invitrogen/Thermo Fisher. The efficiency of each design was tested with serial dilutions of cDNA. PCR reactions (10 µL volume) contained diluted cDNA, 2 × Power SYBR Green Master Mix (Applied Biosystems), 300 nM of forward and reverse primers. PCRs were performed on a SDS 7900 HT instrument (Applied Biosystems) with the following parameters: 50 °C for two minutes, 95 °C for ten minutes, and 45 cycles of 95 °C 15 s–60 °C. Each reaction was performed in three replicates on a 384-well plate. Raw Ct values obtained with SDS 2.2 (Applied Biosystems) were imported in Excel and the normalization factor and fold changes were calculated using the GeNorm method [32].

### 2.5. Hen Egg Chorioallantoic Membrane Model

#### 2.5.1. Ethics Statement

No ethical approval was required for the chick embryo experiments, in accordance with the Swiss animal care guidelines (OPAn, art.112). The experimental design took into consideration the 3R rules in order to improve animal welfare.

#### 2.5.2. Hen Egg Chorioallantoic Membrane Model

Fertilized eggs from white hypex HN hens were obtained at the faculty animal facility (Animalerie d’Arare, University of Geneva, Plan-les-Ouates, Switzerland).

Upon receipt, eggs were carefully wiped with a paper using Milli-Q water then 70% EtOH. They were incubated on Embryonic Development Day 1 (EDD1) for 3 days with the apex down on the automatic rotation mode of the incubator (MG200, Savimat, Chauffry, France) at 37.8 °C under 47% relative humidity. On EDD4, a 3–4 mm hole (in diameter) was drilled in the eggshell at the apex, closed with adhesive tape and incubated apex upwards in stationary mode for 7 days. On EDD11, the hole was enlarged with forceps to a diameter of ~3 cm to allow the treatment and imaging of the chorioallantoic membrane (CAM) vasculature.

#### 2.5.3. Intravenous Administration

On EDD11, a volume of 20 µL of PG3 2/1, PG4 2/1 at 5 and 10 µg or control (WFI) was injected in the main blood vessel of the CAM by mean of a 33-gauge metal needle (30 mm; point style—4; angle—12°; Hamilton) mounted on a 100 µL glass syringe (Gastight #1710, Hamilton). Right after injection, CAM and embryos were imaged with an EOS 90D camera (Canon, Japan) equipped with an EF-S 18–135 mm f/3.5–5.6 IS USM lens (Canon, Japan) and mounted on a tripod. Acquisition parameters were set as follows: 1/100, F5.6, ISO 3200. Eggs were closed with Parafilm^®^ and returned to incubate. Additional images were acquired on days D1, D2 and D3 post injection in order to assess the survival rates of embryos (*n* = 4) over 3 days.

#### 2.5.4. Image Processing

Images were batch cropped to 3000 px * 3000 px around the region of interest (ROI) using FIJI software and a cropping macro (Strock, Christopher. (2021). Batch Cropping Images in ImageJ/Fiji. Zenodo. https://doi.org/10.5281/zenodo.5559119, accessed on 15 July 2022). The ROI was adjusted manually on each image.

### 2.6. Statistical Analysis

The results from multiple quantitative datasets where multiple variables were applied were analyzed using one- or two-way analysis of variance (ANOVA) followed by Tukey′s or Sidak’s multiple comparison test. *p* < 0.05 was considered as statistically significant. * *p* < 0.05; ** *p* < 0.01; *** *p* < 0.001; **** *p* < 0.0001. Data were organized using Microsoft Excel (Microsoft Corporation, Redmond, WA, USA) and statistical calculations were carried out using GraphPad Prism v. 7.0.2 (GraphPad Software, Inc., San Diego, CA, USA). Values are given as mean ± SD.

## 3. Results and Discussion

### 3.1. PAMAM-Glu Synthesis

**PAMAM-Glu synthesis**: Several chemical modifications of PAMAM dendrimers are well-described in the literature [8,10,33,34,35]. PAMAM G3 and G4 were chosen due to their structures and ability to control the number of active amine groups on their surfaces (i.e., 32 and 64 groups for G3 and G4, respectively). Terminal amino groups on PAMAM are crucial for cell transfection, but an excess of these can lead to toxicity. Therefore, PAMAM surface amino groups were blocked by glucuronic acid in order to decrease the toxicity and act as a ligand for the CD206 receptor on M2 macrophages [22]. D-glucuronic acid was also chosen due to its carboxyl group, contrary to other possible CD206 ligands, such as fucose or mannose (Figure 1).

**PAMAM-Glu synthesis:** The water-based, green chemistry reaction for PG3 and PG4 is presented in Figure 1 and Appendix A. In order to protonate only primary amines, EDC and NHS were used at a pH of 5.5 for amidation [36]. Two different degrees of substitution (DS) were tested varying D-glucuronic acid equivalents (0.1 or 0.3 eq.), with the aim to keep enough free dendrimer NH_2_ for ionic interaction with the STING ligand, cGAMP. The DS was calculated from two different complementary methods (i.e., ^1^H NMR spectrum and ninhydrin assay) (Figure 2b). Both PG4 and PG3 0.1 eq. showed comparable DS with both techniques (around 60% coverage) (Figure 2a,b). The DS of PG4 0.3, determined using NMR (i.e., around 80%), was significantly different (*p* < 0.05) to that of ninhydrin (i.e., around 60%). Moreover, the gravimetry results followed the same trend for PG3 0.1 eq. and PG4 0.3 eq., but not PG4 0.1 eq. (Appendix A). However, in all cases, DS were higher (i.e., from two- to six-fold) than expected. These results may be mainly explained by the residual water (Appendix A) and possibly methanol still being present in the lyophilized PAMAM before synthesis, which would decrease the actual amount of PAMAM. Interestingly, the literature has shown similar results in terms of expected free amine groups after the amidation of the amine-terminated PAMAM G3 [36,37,38]. PAMAM dendrimers are known to self-associate, leading to gelation above a critical concentration; we first identified this behavior using rheology and microscopy and then performed gravimetric analyses. The concentrations at which PG3 0.1 eq. and PG4 0.1 eq. were soluble are 0.7 ± 0.26 mg/mL and 0.50 ± 0.12 mg/mL, respectively (Appendix A). Therefore, all the following experimentations were performed below these concentrations, avoiding aggregation. Finally, affinity towards CD206 was measured by MST, showing that PG3 has the highest affinity among the three polymers tested (Appendix A). Moreover, its Kd (around 3 × 10^7^ nM) value was significantly lower compared to PG4 0.3 eq. (*p* < 0.01).

Due to the better balance between blocked and free amino groups, offering sufficient binding sites for STING ligand–cGAMP complexation, a 0.1 equivalence was selected for further investigations. Moreover, PG3 0.1 eq. showed the highest CD206 affinity out of the three dendrimers examined, and consistent DS values with three measurement methods. Still, further investigations were carried out with both PG3 and PG4 0.1 eq. and all further preparations were prepared below their gelation concentrations.

### 3.2. cGAMP Complexation with PG3/PG4

**PAMAM-Glu complexation with cGAMP**: Once PAMAM-Glu was synthesized, characterized and its reproducibility was assured, the next step was to complex it with the negatively charged cGAMP. Nanocomplexes (NCs) were prepared following the ionic gelation method at two different molar charge ratios (N/P), 1/1 and 2/1 (Table 2, Figure 3). NCs were analyzed by AF4/UV/RI/MALS (Figure 4a–c) and their complexation efficiencies were compared to UPLC/UV (Figure 4c).

**AF4 RI/UV and MALS analysis of NCs:** The AF4 method was employed in order to evaluate different NC populations. We observed only one population using three different detectors: UV, RI and MALS (Figure 4 and Appendix A). Moreover, based on the UV concentration detector, we obtained recovery data corresponding to the complexation of cGAMP with PG3/PG4. We observed that the RI signal originated mostly from the dendrimer, as cGAMP alone does not generatea strong signal [39], while for the UV signal we found that the opposite was true, corresponding specifically to cGAMP absorption at 256 nm (Figure 4a,b and Appendix A) [39]. MALS (90°) represents both cGAMP and PG3/4 in NCs (1/1 and 2/1). It can be observed that RI values are comparable for both PG3 and PG4 in NCs (Appendix A). Due to the fact they had the same elution time, we can speculate that cGAMP complexed with PG3 or PG4. Finally, the UV signal in NC was higher, especially in the case of PG4 2/1 compared to the rest of formulations (Figure 4a,b).

**Complexation efficacy (CE% by UPLC/UV) and recovery (AF4/UV) of NCs were compared:** CE% was around 20% and recovery was 10% for all NCs (Figure 4c). Interestingly, the same difference (recovery being two times lower compared to CE%) was observed in one of our previous works with a different, positively charged polymeric system [39]. This difference might arise from NCs sticking to the membrane, or as a result of strong cross flow forces (AF4/UV), separating loosely bound PG and cGAMP in NC. Complexation efficiency was significantly different compared to recovery for the 1/1 ratio (both PG3 and PG4), suggesting less stable 1/1 complexation. 

Two different PAMAM generations, G3 and G4, bound to glucuronic acid (PG3 and PG4) were complexed with cGAMP (Table 2; Figure 3). As a comparison, Bono et al. worked on PAMAM/pDNA, and prepared NCs at 1/1 and 2/1 ratios. They showed that decorated PAMAM for lower (G2) and medium (G4) generations exhibited maximal complexation ability with pDNA at N/P ≤ 2 [12]. Additionally, they reported that for N/P ≥ 5, most of the dendrimer was not engaged in interactions with pDNA and consequently provoked higher cytotoxicity. Additionally, due to the limitation explained above, high concentrations of PAMAM would cause gelation, and to avoid this we selected ratios of 2/1, as the test, and 1/1, as the control. We showed that NCs contained 10–20% encapsulated cGAMP and that, owing to the same elution time observed with RI or UV detection, we hypothesized that complexation was successful. G3 PAMAM, having a lower number of branches, is reported in the literature to be more flexible than G4. Thus, G3 is more likely to incorporate small molecules such as cGAMP within its core, but is more prone to instabilities such as aggregation [8,35]. To examine the physicochemical properties of both PG3 and PG4 complexed with cGAMP in depth, in we investigated the size, charge and shape of NCs, employing different orthogonal techniques.

It is well-acknowledged that size, charge and shape are among the main aspects affecting the transfection performances of gene delivery systems, at least in vitro [40,41]. Therefore, the aim of this work was to examine these parameters and investigate differences between different PAMAM generations (Figure 5). Moreover, to understand each of the parameters, at least two orthogonal methods were employed. Thus, indirectly, complexation between PAMAM-Glu and cGAMP was investigated.

**Size of NCs:** NTA, as a number-based particle-by-particle method, was compared to an intensity-based (z-average) technique—DLS. DLS is sensitive towards big particles, while NTA gives the mean value of the whole population. If the sample is monodisperse, the intensity-based value would overlap with the number-based technique. In this work, dendrimers alone, both PG3 and PG4 controls, show significantly different NTA sizes compared to DLS (*p* < 0.01 for PG3 and *p* < 0.001 for PG4) and AF4-DLS (*p* < 0.0001), suggesting polydisperse and colloidally instable PG3 and PG4 samples (Figure 5a). However, the addition of cGAMP stabilizes the formulation, suggesting a possible complexation, as supported by:-The absence of significant differences between the NTA, DLS and AF4-DLS for NCs (PG3 150–200 nm and PG4 200–270 nm) (Figure 5a);-PG NCs being significantly smaller by DLS compared to PG alone, especially in the case of the 2/1 ratio (PG3 2/1 < PG3 *p* < 0.0001; PG4 2/1 < PG4 *p* < 0.01).

**NC aggregation analysis**: After AF4-DLS analysis, within the only peak eluted, PG3 NCs had a Dh (2 × Rh) almost equal to that of DLS—around 200 nm—while PG4 NCs were in range of 80–100 nm, which is smaller than their DLS and NTA sizes, especially in the case of 2/1 PG4 (*p* < 0.01), for which the polydispersity of the sample and possible aggregation might be the reason (Figure 5a).

**NCs′ zeta potential**: Aiming for an efficient transfection, as well as targeting ability, charge plays an important role. As the cell membrane is negatively charged, zeta potential analysis was performed (Figure 5b). When positively charged PAMAM such as PG3 (around 25 mV) and PG4 (around 10 mV) were complexed with negatively charged cGAMP, their surface potentials and thus charges were expected to decrease. This was confirmed in the case of PG3, having both PG3 1/1 and 2/1 zeta potentials significantly lower compared to the PG3 alone (*p* < 0.0001). Moreover, the pH of the PG3 2/1 suspension is (pH 6) most similar to the pH of the TME. For all NCs, zeta potential was significantly different between DLS and NTA (*p* < 0.01) (Appendix A).

**NCs′ shapes:** By employing AF4-MALS-DLS, we obtained Rg (gyration radius—representing mass distribution) and Rh (hydrodynamic radius) values, and the so-called shape factor Rg/Rh. Based on shape factor, NCs can be classified as micro(nano)gels for values <0.7; homogeneous spheres for 0.775; branched molecules (1–1.5); random coils (1.5–2.1); and rod-like, elongated structures for values >2 [42]. Based on their shape factors, 1.4 ± 0.3 for 1/1 and 1.2 ± 0.5 for 2/1, PG3 NCs resemble smaller-branched molecules compared to PG3 alone, which has a higher shape factor. A similar trend was observed for PG4 NCs (Figure 5c). However, PG4, PG4 NCs and PG3′s shape factors were out of range, higher than 10, probably due to the polydispersity of the samples. Our investigation using SEM obtained results that are in accordance with the AF4-MALS-DLS findings, where we can clearly see the difference between PG3 and PG4 compared to NCs (Figure 5d and Appendix A).

Considering size, charge, shape and complexation efficiency together, we may pre-select the most adequate(s) nanocomplexes. Besides the grafting of specific targeting moieties (D-Glu—CD206 Rc), we aimed for a particle size of around 200 nm to ensure efficient macrophage phagocytosis [43]. We observed a mean DLS size of 150–200 nm for PG3 NCs and 200–270 nm for PG4 NCs, having a PG3 zeta potential significantly higher compared to PG4. The same size and charge difference were already shown by Bono et al., who found that G4 was slightly larger and less charged compared to G3 once complexed with pDNA [12]. We indirectly showed that both PG3 and PG4 were complexed with cGAMP, as confirmed by their smaller sizes, monodispersed distributions compared to the more polydisperse free dendrimers, and significantly lower zeta potentials. Moreover, this difference was confirmed by SEM. NCs observed by SEM were smaller compared to DLS and NTA, as expected due to the desolvation of PAMAM [34]. Overall, PG3 seems to be more stable compared to PG4 based on its size, charge, shape and CE status properties. However, in vitro and in vivo studies are needed for both dendrimers to decide on the optimal complexation ratio and polymer.

### 3.3. In Vitro Toxicity and Efficacy of NCs

**NCs do not cause toxicity:** To investigate whether NCs can have direct cytotoxic effects on macrophages, we cultured M1 and M2 polarized BMDMs in 1% and 21% O_2_ with cGAMP, PG3 (1/1 or 2/1) or PG4 (1/1 or 2/1) NCs (Figure 6). After 24 h, the percentage of live cells was measured by flow cytometry with live/dead assay. 

The results indicated that NCs did not show obvious (*p* > 0.99) toxic effects against M2 or M1 polarized BMDMs under normoxic (21% O_2_) conditions (Figure 6a). Furthermore, BMDMs cultured in hypoxic conditions (1% O_2_) in order to mimic TME milieu were not significantly (*p* > 0.99) affected by the NCs compared to the control group (Figure 6b). Therefore, macrophage treatment with formulated NC with cGAMP has a minimal or no effect on cell viability.

TAMs can display a continuum of phenotypes manifested by distinct gene expression profiles with the capacity to switch from one phenotype to another depending on the external signaling. In cancer, most observations related to the TAM phenotype switch are linked to the mechanisms of escape from immunological surveillance [44]. Thus, TAM polarization is currently considered to be an important mechanism of immune escape growth control in cancer [44,45]. Therefore, an ideal approach to target tumor-infiltrating macrophages is not through depleting them but rather converting M2-like TAMs into M1 anti-tumor macrophages. The activation of the cGAS-STING signaling pathway in macrophages is shown to affect polarization, which is necessary for coupling innate and adaptive immune responses, thereby regulating tumor progression [46,47].

**NC treatment of BMDMs is associated with changes in surface marker expression.** In order to investigate the efficacy of NC to deliver cGAMP and thereby impact macrophage polarization, we used in vitro polarized M1 or M2 BMDMs that were treated with NC for 24 h and cultured in standard 21% O_2_ conditions or 1% O_2_. The surface expression of the macrophage differentiation markers, CD68, the prototypical M1 marker, CD80 and CD86, and the M2 marker, CD206, was then analyzed by flow cytometry to confirm polarization. The results showed that CD86 expression in M2 cells was significantly enhanced after 24 h of NC incubation in normoxic conditions and especially in hypoxic conditions (Figure 7a,b). This was also observed in cells treated with cGAMP alone, suggesting that NC can successfully repolarize BMDMs toward an M1-like phenotype in a similar manner as cGAMP. This effect was not observed in M1 macrophages since their polarization status was already strongly induced (IFN-γ and LPS stimulation). However, we did not detect a significant effect of NCs on the expression levels of CD80, CD206 and CD68 in both M1 and M2 BMDMs (Figure 7 and Appendix A). These results emphasize the benefit of using NC in the targeted delivery of cGAMP or other STING agonists and thereby overcoming the dose-limiting toxicities that could occur as a result of systemic delivery [48]. Indeed, the increased CD86 expression by cGAMP-formulated NCs can provide the costimulatory signals necessary for T cell activation and survival by binding to CD28 or/and cytotoxic T-lymphocyte-associated protein 4 (CTLA-4).

NC upregulated the mRNA expression of proinflammatory (M1) genes and reduced the expression of the M2-associated gene *Mrc1* in BMDMs. To further investigate changes in the status of BMDMs after treatment with NC, we used two gene signature panels that enabled us to assess the polarization status of the BMDMs based on their mRNA expression profiles. For BMDMs undergoing classical activation (M1), we used *Il1b*, *Nos2* and *Stat1* gene expression signatures (Figure 8a and Appendix A). For alternatively activated BMDMs, we measured standard M2 markers, *Mrc1* and *Arg1* (Figure 8b and Appendix A [14,49]. Among the upregulated genes in M2 macrophages in vitro is mannose receptor *Mrc1*, also known as CD206. *Mrc1* expression promotes the expression of several anti-inflammatory cytokines and chemokines such as TGF Β, IL10, and CCL18 [15]. Indeed, cGAMP and NC treatment reduced the expression of *Mrc1* in M2 macrophages (Figure 8a) as well as in M1 macrophages, Appendix A. Surprisingly, the expression of *Arg 1* was increased in cGAMP and NC-treated M2 macrophages, in contrast to the lower expression level seen after the treatment of M1 macrophages (Appendix A). This might reflect heterogeneity within the treated M2 macrophages, or incomplete repolarization at the early time-point tested. To assess the effect of NC on M1-specific gene signature, we measured the expression levels of *Il1b*, *Nos2* and *Stat1* [50]. Clearly, the mRNA levels of the genes in this signature showed that cGAMP and NCs increased the expression of M1-specific markers in M1 and M2 BMDMs. In particular, PG3 2/1 showed a very noticeable effect increasing the expression of *Nos1* and *Stat1* genes and downregulating the expression of *Mrc1* gene in both M1 and M2 polarized BMDMs (Figure 8d,e). The stimulation of the cGAS/STING pathway promotes anti-tumor immunity through mediating type I interferon (IFN) production [51]. To test if cGAMP and formulated NC could induce type I IFN secretion, we measured the cytokine levels in the supernatant of the BMDMs. Indeed, cGAMP induced IFN-β secretion in M1 and M2 BMDMs compared to the control. The same was observed in BMDMs treated with NC (especially in the case of PG3 2/1), which were able to induce IFN-β particularly in M2 macrophages at 50% compared to control (Appendix A).

### 3.4. In Vivo Toxicity of NCs

**Biocompatibility of NCs in chicken embryos:** Considering that the 2/1 ratio was more effective compared to 1/1, in terms of its ability to control macrophage phenotype from M2 to M1, especially in the case of PG3, our final experiments used the 2/1 ratio. However, the last biocompatibility examination was performed before deciding which dendrimer will be the carrier of choice (Figure 9).

Toxicity was assessed by observing the embryo survival (live or dead) at 1 min and 1, 2 and 3 days after an injection of 20 µL of WFI, 5 µg or 10 µg of PG3 2/1 or PG4 2/1 (Figure 9a). We observed no obvious toxicity in embryos treated with both doses of PG3 2/1 compared to WFI over time (Figure 9b). Similarly, PG4 2/1 did not induce embryo death at anytime, however CAMs treated with this compound manifested whitish structures around the injection site at 1 day, that spread along the vein at 2 and 3 days for both 5 µg and 10 µg doses (Figure 9b). It might come from the fact that even at lower doses, PG4 tends to be more viscous and induce lower pH compared to PG3 (Appendix A).

Considering that PG3 2/1 did not express any unusual phenomena at the site of injection, PG3 2/1 blood concentrations on injection day (EDD11) were estimated to be 3.731 and 7.463 µg/mL for 5 and 10 µg doses, respectively, and considered biocompatible and non-toxic [52].

## 4. Conclusions

The results obtained in this work show that PG is a promising cGAMP transfection carrier. The characterization process provided insightful information about the polymer’s behavior in solution. The aggregation of PG3 and PG4 at a high concentration was increased, as confirmed by the increase in viscosity found using rheological measurements and from light microscopy observations. It is therefore important to prepare PG solutions below the gelation concentration. Complexation between PG and cGAMP was indirectly proven by size and zeta potential measurements, where particles were smaller and less charged compared to empty NCs. Moreover, AF4-UV and UPLC-UV showed 10 and 20% cGAMP recovery and complexation efficiency in NCs, respectively. Based on the in-depth orthogonal NC characterization by DLS, NTA, SEM, and AF4-MALS-DLS, the PG3 NCs seemed to be more stable and monodispersed, showing less aggregation and larger particles (200 nm) compared to PG4 NCs. Additionally, the PG3 carrier showed a higher affinity towards CD206, and the PG3 2/1 NCs demonstrated a clear effect increasing the expression of NOS1 and STAT1 genes characteristic for M1 macrophages, downregulating the expression of ARG1 and MRC1, characteristic for M2 genes and increasing IFN-β cytokine in SN in both M1 and M2 polarized BMDMs. Finally, PG3 2/1 was the condition that caused the fewest side effects on chicken embryos at both examined doses (5 and 10 µg). This work shows that decorating PAMAM with CD206 ligand such as glucuronic acid has potential as a viable M2 targeting transfection carrier for local cGAMP delivery to the tumor microenvironment of cold tumors.

## Figures and Tables

**Figure 1 pharmaceutics-14-01883-f001:**
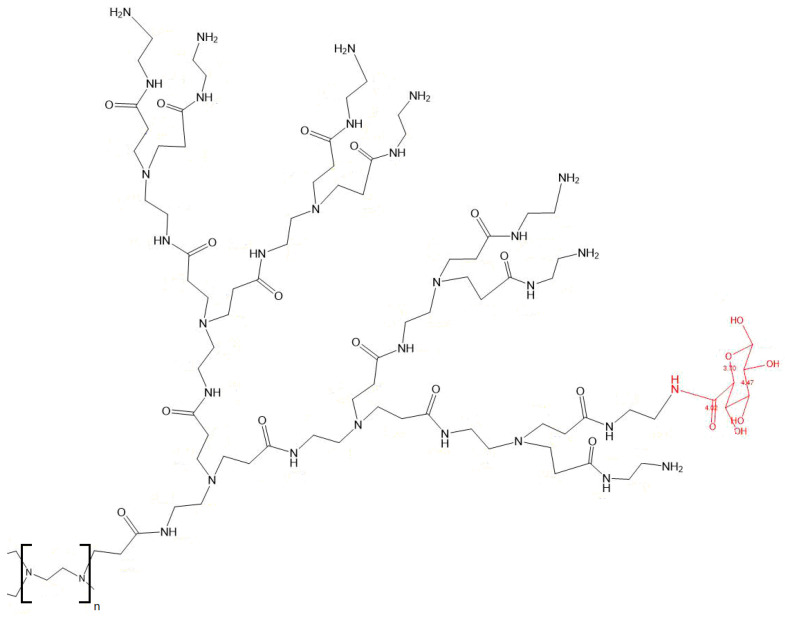
**Chemical structure of a branch of PAMAM D-Glucuronic acid**.

**Figure 2 pharmaceutics-14-01883-f002:**
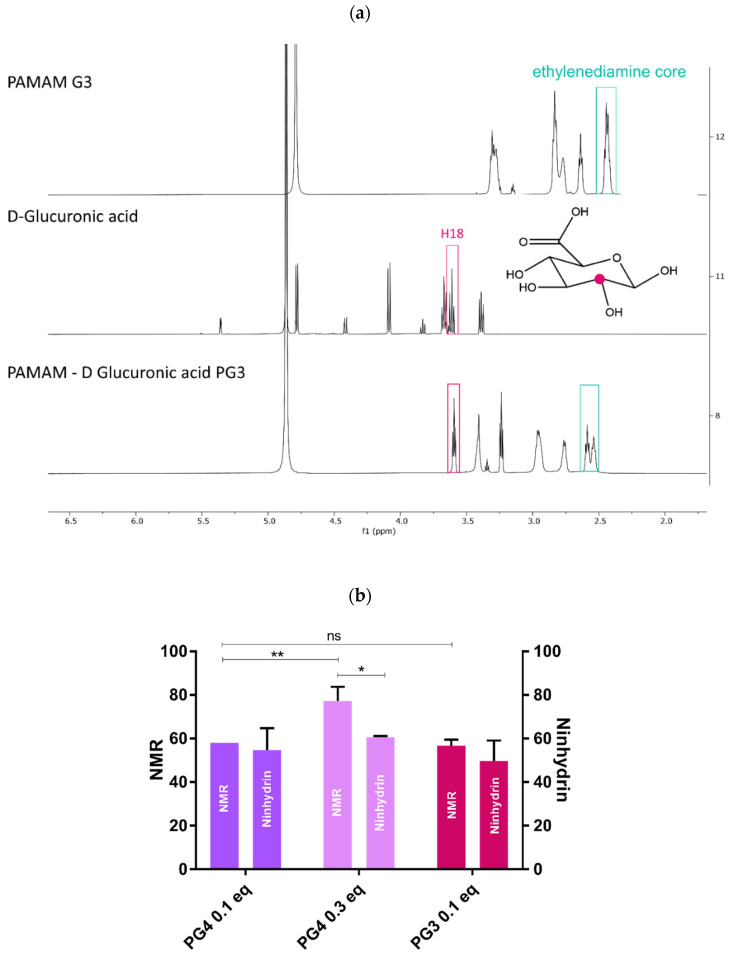
**PAMAM G3 grafted through D-Glucuronic acid (PG3 0.1 eq.) synthesis and characterization:** (**a**) NMR peak representation; (**b**) NMR and ninhydrin assay comparison of degree of substitution (DS)/grafting between PG3 0.1 eq., PG4 0.1 and 0.3 eq. * *p* < 0.5; ** *p* < 0.01; ns—non significant.

**Figure 3 pharmaceutics-14-01883-f003:**
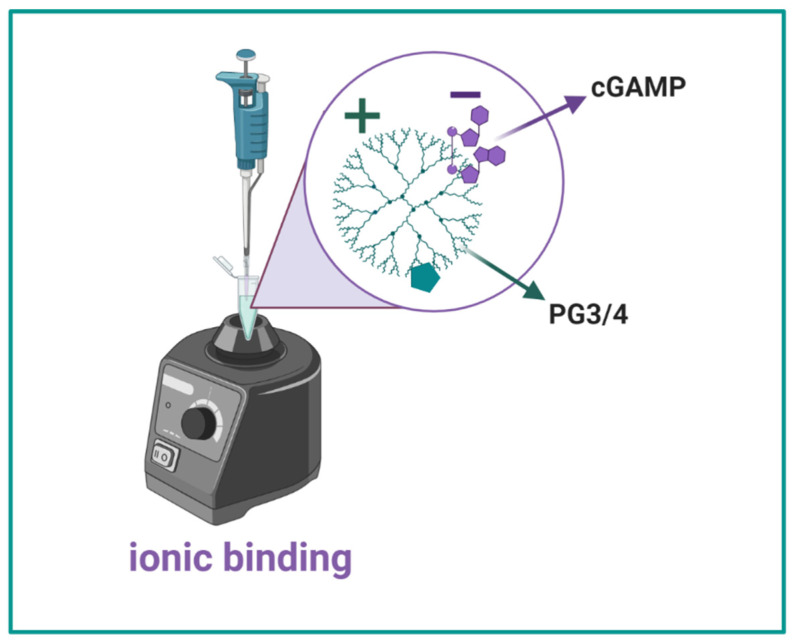
**PG/cGAMP 1/1 and 2/1 NC**: Ionic binding between positively charged PG and negatively charged cGAMP while vortexing for 30 s (RT at pH 6) at molar charge N/P ratios of 1/1 and 2/1.

**Figure 4 pharmaceutics-14-01883-f004:**
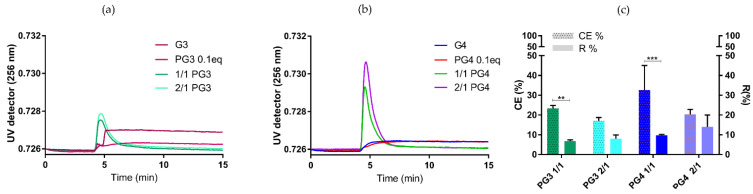
**Complexation efficacy and recovery measured by UPLC-UV and AF4-UV** (**a**,**b**) AF4-UV signal of different NCs; (**c**) complexation efficiency and recovery comparison of cGAMP attached to PG. ** *p* < 0.001; *** *p* < 0.001.

**Figure 5 pharmaceutics-14-01883-f005:**
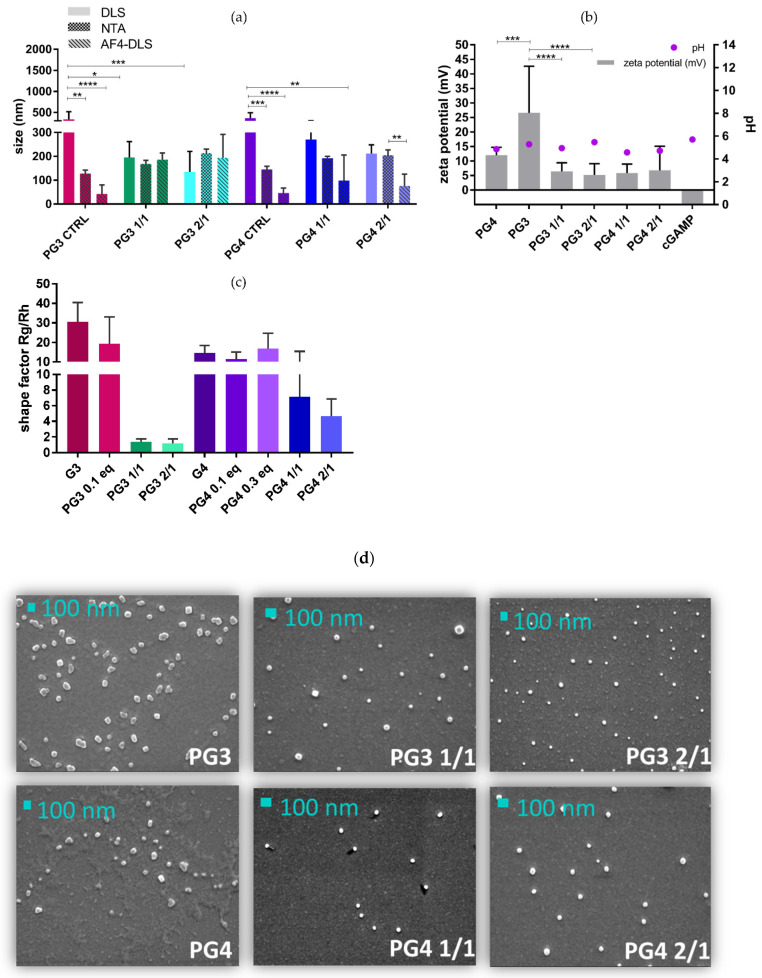
**NC size, charge, and shape comparison at ratios of 1/1 and 2/1**. (**a**) Size comparison through DLS, NTA and AF4-DLS; (**b**) DLS: NC charge; (**c**) AF4-MALS-DLS: shape factor gyration (Rg)/hydrodynamic radius (Rh); (**d**) SEM: shape of the NCs. * *p* < 0.05; ** *p* < 0.01; *** *p* < 0.001; **** *p* < 0.0001.

**Figure 6 pharmaceutics-14-01883-f006:**
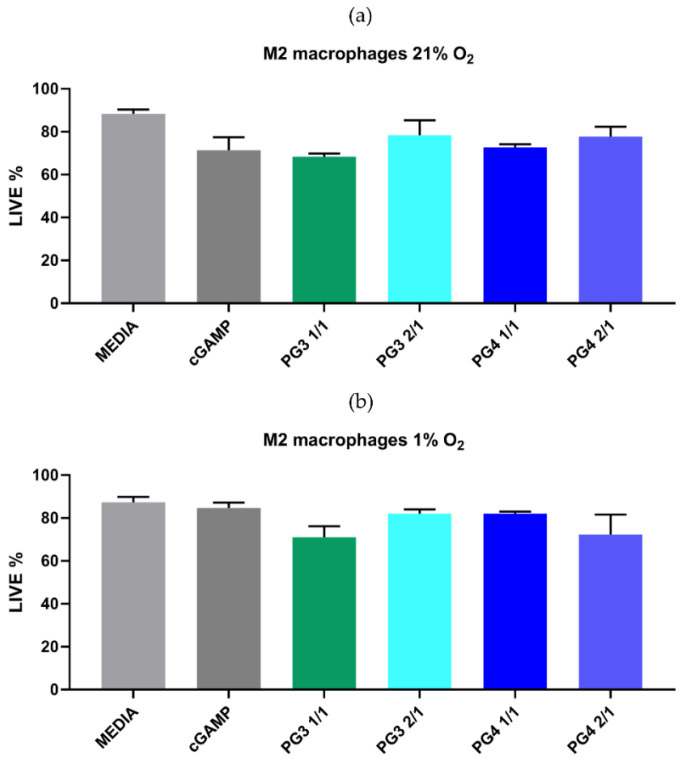
**PAMAM NCs do not affect M2 macrophage viability under normoxic and hypoxic conditions**. BMDM cell viability was determined by live/dead assays. The polarized M2 BMDMs were cultured in normoxic (21% O_2_) (**a**) or hypoxic (1% O_2_) (**b**) conditions and treated with corresponding PAMAM NCs (PG3 1/1, PG3 2/1, PG4 1/1 or PG4 2/1), cGAMP (25 µg/condition) or with media. After 24 h, cells were collected stained using a Live/Dead Viability Assay Kit to evaluate cell viability and analyzed by flow cytometry.

**Figure 7 pharmaceutics-14-01883-f007:**
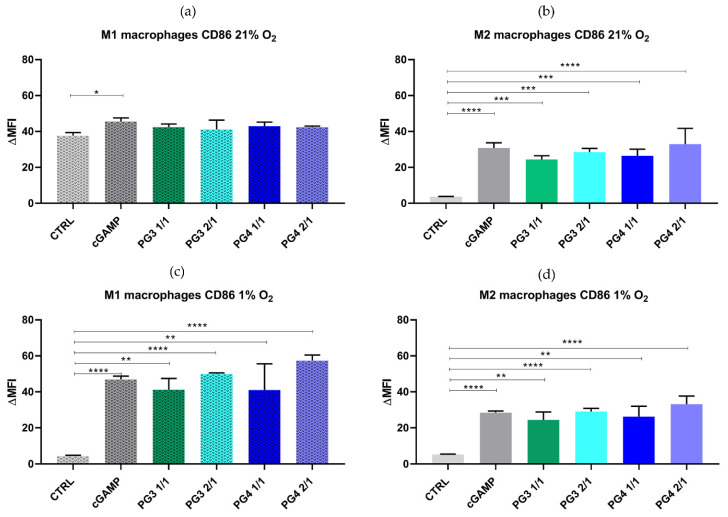
**PAMAM upregulates surface expression of costimulatory molecule CD86 on murine BMDMs**. Polarized M1 and M2 BMDMs were stimulated in vitro with corresponding PAMAM NCs (PG3 1/1, PG3 2/1, PG4 1/1 or PG4 2/1), cGAMP (25 µg/condition) or with media under normoxic (21% O_2_: (**a**,**b**)) or hypoxic (1% O_2_: (**c**,**d**)) conditions for 24 h. The cells were stained with fluorophore-conjugated antibodies against CD86 and analyzed by flow cytometry. Error bars are SEM of three independent experiments. * *p* < 0.05; ** *p* < 0.01; *** *p* < 0.001; **** *p* < 0.0001.

**Figure 8 pharmaceutics-14-01883-f008:**
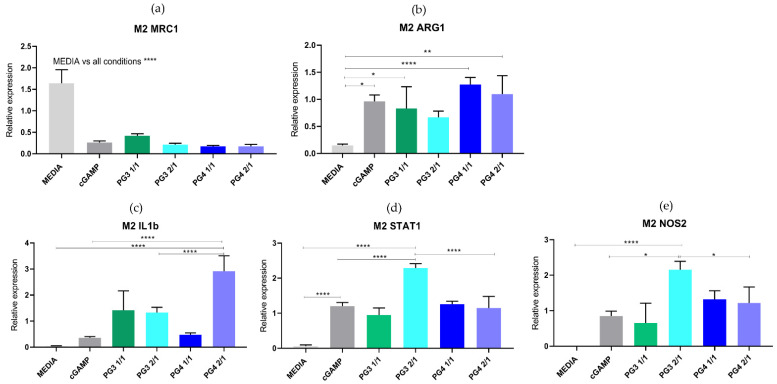
**NCs upregulate mRNA expression of pro-inflammatory (M1) genes and reduce the expression of the M2-associated gene *Mrc1* in M2 polarized BMDMs.** Relative mRNA expression of M2-like macrophage markers (*Mrc1* (**a**) and *Arg1* (**b**)) and of classically activated (M1-like) macrophage markers (*IL-1b* (**c**), *STAT1* (**d**) and *NOS2* (**e**)) in M2 BMDMs treated with corresponding PAMAM NCs (PG3 1/1, PG3 2/1, PG4 1/1 or PG4 2/1), cGAMP (25 µg/condition) or with media. * *p* < 0.05; ** *p* < 0.01; **** *p* < 0.0001.

**Figure 9 pharmaceutics-14-01883-f009:**
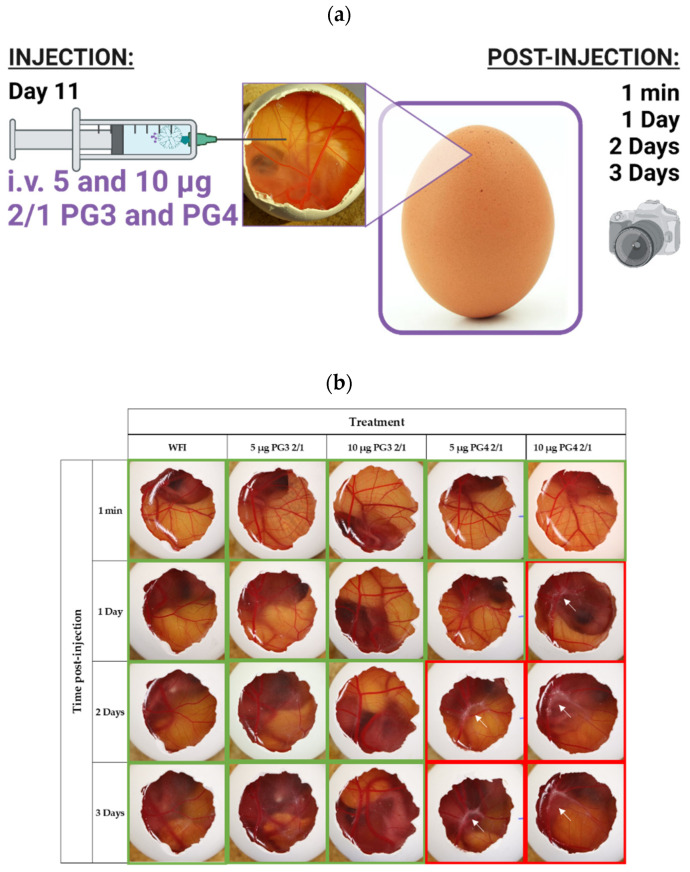
**In vivo examination of NC toxicity in chicken embryos**. (**a**) graphical representation of the experimental procedure (**b**) Representative images of hen egg CAMs injected intravenously with 20 µL of: WFI, 5 µg or 10 µg of 2/1 PG3 or 2/1 PG4. Pictures were acquired at 1 min, 1 day, 2 days or 3 days post-injection. At each time point, the viability of the embryos was confirmed by observing the integrity of the CAM and the embryos’ spontaneous movements. *n* = 4; eggs in the figure were selected as ¾ eggs having the same phenomena. Green frames correspond to normal and red to abnormal behavior, pointed by white arrows.

**Table 1 pharmaceutics-14-01883-t001:** Parameters for AF4 method: Detector flow was kept at 0.5 mL/min cGAMP recovery (%) Rg, Rh, MW and values for RI and UV intensity on the graphs are represented as the mean value of three different batches of PG3 and PG4.

FOCUS STEP	TIP flow	0.20 mL/min	3 min
	Cross flow	1 mL/min	3 min
	Focus flow	1.30 mL/min	0.2 min
ELUTION STEP	Constant cross flow	1 mL/min	0.2 min
	Exponential cross flow power 0.2	1–0.1 mL/min	10 min
	Linear cross flow	0.1–0 mL/min	10 min
RINSE STEP	TIP flow	0.1 mL/min	0.5 min
	Focus flow	0.1 mL/min	0.5 min

**Table 2 pharmaceutics-14-01883-t002:** Conditions.

Controls			PG/cGAMP = Nanocomplexes (NCs)	
			1/1 or 2/1 PG3/cGAMP	1/1 or 2/1 PG4/cGAMP
**cGAMP**	**PG3**	**PG4**	**PG3 1/1 or 2/1**	**PG4 1/1 or 2/1**

## Data Availability

The data presented in this study are available on request from the corresponding author.

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
