# Peer review of "Synthesis, Formulation and Characterization of Immunotherapeutic Glycosylated Dendrimer/cGAMP Complexes for CD206 Targeted Delivery to M2 Macrophages in Cold Tumors"

_pharmaceutics, 2022, doi:10.3390/pharmaceutics14091883_

Round 1

Reviewer 1 Report

The manuscript submitted by Petrovic et al. investigates the transformation of M2-type pro-tumoral macrophages into M1-type anti-tumoral macrophages upon PAMAM-mediated intracellular delivery of cyclic guanosine monophosphate–adenosine monophosphate (cGMAP), where the CD206 receptors of the macrophages are targeted by glucuronic acid.

The experimental section of the manuscript quite rich. Characterisation and analysis of the PAMAM-cGMAP-glucuronic acid complexes by various techniques increases the significance of the manuscript.

The abstract of the manuscript is informative and intriguing, yet the introduction section does not catch up with the abstract. Particularly, the lines 102-119 of the introduction section had been written superficially in comparison to rest of the introduction section. Introduction section must be improved after line 102. APC targeting is obviously crucial to support the hypothesis, yet the section describing the APC targeting is not well-integrated with the rest of the introduction section. Similarly, the lines 114-119, where the main goal of the research is described, are too brief and not adequately justifying the context of the research.

Notably, the in vivo experiments carried out on hen egg is a very good choice to study the impact of PAMAM-mediated delivery. However, the information provided about the integrity of CAM and embryo survival does not provide enough insights for assessing the effectiveness of PAMAM-mediated cGMAP delivery for macrophages transformation. The images provided in figure 9 are beautiful yet they do not really provide much information (they are very similar too each other for non-trained eyes). Information provided on blood concentrations do not really say much about the physiological condition of the embryos. Providing a brief analysis on the biochemical profile of the blood samples would be much more informative. Probably, a more detailed approach by transplanting the tumor cells to the egg would provide the most invaluable information (please see https://doi.org/10.1038/s41598-018-25573-8).

Generally speaking, I do not find the manuscript unsuitable for publication. I believe that improving the in vivo experiments would highly increase the impact of the manuscript. The detailed physiochemical analysis on the PAMAM nano-complexes had been well-presented and very valuable. However, the manuscript requires certain improvements before publication.

Author Response

see attached file

"pharmaceutics-1838722_Answer to reviewers 1 comments.docx"

Reviewer 2 Report

The manuscript titled "Synthesis, formulation and characterization of immunotherapeutic glycosylated dendrimer/cGAMP complexes for CD206 targeted delivery to M2 macrophages in cold tumours"  contains a well-designed research plan and the outcomes of the research are very encouraging.

Overall, the manuscript was very well written, and experimental procedures and outcomes are provided with a logical explanation and references. 

The suitable N/P ratio remains and requires further investigations for additional in vivo and invitro experiments.

There are a few typographical errors (protocols) and other parts of manuscript required to be addressed

The manuscript titled "Synthesis, formulation and characterization of immunotherapeutic glycosylated dendrimer/cGAMP complexes for CD206 targeted delivery to M2 macrophages in cold tumours"  contains a well-designed research plan and the outcomes of the research are very encouraging.

Overall, the manuscript was very well written, and experimental procedures and outcomes are provided with a logical explanation and references. 

The suitable N/P ratio remains and requires further investigations for additional in vivo and invitro experiments.

There are a few typographical errors (protocols) and other parts of manuscript required to be addressed.

Author Response

see attached file

"pharmaceutics-1838722_Answer to reviewers 2 comments.docx"

Reviewer 3 Report

The article refers to the preparation of a series of modified PAMAM dendrimers with D-glucuronic acid and 2`3` Cyclic GMP-AMP, and the evaluation of their influence in the expression of genes characteristic for M1 macrophages.

Although the topic is of interest, the manuscript lacks important aspects regarding the characterization of the synthetized compounds, and for this reason, mayor revisions are needed.

Regarding the synthesis and characterization of the manuscript:

- In the “2. Materials and Methods” sections, in “2.1.Chemicals”, two different PAMAM with an ethylenediamine core G4 were purchased from Sigma-Aldrich:  4th (G4) generations (10 wt% solutions in methanol) and PAMAM G4 (G4) with an ethylenedia-mine core (10 wt% solution in methanol). Author must explain why is this difference important?

- In the “2.2 Synthesis of G3 and G4 PAMAM-D-glucuronic acid derivatives” is written: D-Glucuronic (0.1 eq. NH2 for PG3 0.1 eq and PG4 0.1 eq, and 0.3 eq for PG4 0.3 eq)), this is confusing and must be clarified.

- If only 0.1 eq. NH2 for PG3 of D-Glucuronic is used, I do not understand why the coverage is 60%. Authors must explain it!! Do you use 0.1 eq. Per –NH2 group? Or per PAMAM? If you used only 0.1 eq. per –NH2 group is not expected such coverage degree!!!

- In figure 1, the structure of PAMAM dendrimers is too small and con be clearly distinguished. Additionally, authors must add the chemical structure of the formed compound (PG3 or PG4).

- Too much information about the degree of substitution (DS) of the dendrimers is obtained from NMR integrations. However, no integrated spectra appear nor in the manuscript or in the ESI. If the information about DS is maintained, then authors should include all integrated NMR spectra in the ESI and all calculations done to reach this results.

- In figure 2, NMR are not clear. Is the signal around 5.5 ppm in the D-Glucuronic spectrum really the carboxylic proton? If the spectrum is made in D2O, should not be this proton deuterium due to solvent exchange?

Also in figure 2, NMR spectrum of PG3 is not clear. It seems that there is not D-Glucuronic in the structure. Authors must explain this with a better NMR characterization to be completely sure about the existence of PG3.

- All the experiments for PG4 should be also included.

Author Response

see attached file

"pharmaceutics-1838722_Answer to reviewers 3 comments.docx"

Reviewer 4 Report

In this paper the authors described the development and characterization of PANAN dendrimer loaded with cGAMP to reverse macrophage polarization from M2 to M1.

This work is important and particle characterization and testing was convincing enough to induce the reader to believe that this strategy can be effective.

-I did not like the presentation of the data. I suggest to put together more figures in panels and to leave in SI what is really supplementary.  Figure 4 for example is very hard to evaluate. The colors used in 4b for examples represent the worst combination to chose to highlight a difference. The sequence of the analysis also is deceptive alternating LS90, RI and UV detection in a weird way. In addition, the authors should spend more words for explaining the data.

-Figure 8: can the author explain the trend f Arg1? Despite what is claimed in the text , its expression increases.

-Can the check in the particle can activate NFkB to reverse m2 to M1?

-The author evaluated the targeting of cd206 through a chemical way, but when we talk about particles they should perform FACS analysis and show an increased internalization at least in comparison with untargeted particles.

-The authors did not follow the journal guidelines. The paper is missing the discussion and they do not cite very similar work in literature. They need to state how this paper is different from other work in the field aiming at the same goal.

I think with some substantial modifications this paper is worth for publication.

Author Response

see attached file

"pharmaceutics-1838722_Answer to reviewers 4 comments.docx"

Round 2

Reviewer 1 Report

I appreciate the revised the version of the manuscript with the changes introduced by the Authors. However, I still do not find the in vivo section satisfactory. In my previous comments, I have pointed out that the biochemical profile of the blood samples would be more informative. It seems the Authors skipped this point.

The toxicity of the nano-complexes were concluded by referring to blood concentrations. However, I cannot understand how the toxic effects can be concluded by the changes in blood concentrations. I am not entirely convinced by the pervious study from 1975 that was cited by the Authors regarding the blood concentrations. I find neither the color changes in the veins nor the blood concentrations conclusive about toxicity.

I kindly ask Authors to provide more information about the blood sampling techniques and methods used for the measurement of blood concentrations. More importantly, I ask Authors to write down what happened to the embryos after the experiments (dead or alive?). If the embryos were dead, could not Authors extract the blood from embryos and provide a full biochemical profile of the blood samples to define the toxic effects of the nano-complexes? If the embryos were alive, couldn’t the Authors wait couple of days more to extract blood samples from pre-hatched or fully hatched chicks for a detailed biochemical analysis?

I think that more discussion on the in vivo studies are required before the manuscript is accepted for publication. I certainly recommend Authors to consider full biochemical analysis before revising the manuscript.

Reviewer 3 Report

In my opinion, the manuscript in its present form, has been significantly improved and is suitable for publication.

Reviewer 4 Report

N.A